# Polarization Performance Simulation for the GeoXO Atmospheric Composition Instrument: $NO_2$ Retrieval Impacts

Aaron Pearlman[1,3], Monica Cook[1,3], Boryana Efremova[1,3], Francis Padula[1,3], Lok Lamsal[2,3], Joel McCorkel[3], and Joanna Joiner[3]

[1]GeoThinkTank LLC, Miami, FL, USA
[2]University of Maryland Baltimore County (UMBC), Baltimore, MD, USA
[3]NASA Goddard Space Flight Center, Greenbelt, MD, USA

**Correspondence:** Aaron Pearlman (aaron@geothinktank.com)

**Abstract.** NOAA's Geostationary Extended Observations (GeoXO) constellation will continue and expand on the capabilities of the current generation of geostationary satellite systems to support US weather, ocean, atmosphere, and climate operations. It is planned to consist of a dedicated atmospheric composition instrument (ACX) to support air quality forecasting and monitoring by providing similar capabilities to missions such as TEMPO (Tropospheric Emission: Monitoring Pollution), currently planned to launch in 2023, and OMI (Ozone Monitoring Instrument), TROPOMI (TROPOspheric Monitoring Instrument), and GEMS (Geostationary Environment Monitoring Spectrometer) currently in operation. As the early phases of ACX development are progressing, design trade-offs are being considered to understand the relationship between instrument design choices and trace gas retrieval impacts. Some of these choices will affect the instrument polarization sensitivity (PS), which can have radiometric impacts on environmental satellite observations. We conducted a study to investigate how such radiometric impacts can affect $NO_2$ retrievals by exploring their sensitivities to time of day, location, and scene type with an ACX instrument model that incorporates PS. The study addresses the basic steps of operational $NO_2$ retrievals: the spectral fitting step and the conversion of slant column to vertical column via the air mass factor (AMF). The spectral fitting step was performed by generating at-sensor radiance from a clear sky scene with a known $NO_2$ amount, the application of an instrument model including both instrument PS and noise, and a physical retrieval. The spectral fitting step was found to mitigate the impacts of instrument PS. The AMF-related step was considered for clear sky and partially cloudy scenes, where instrument PS can lead to errors in interpreting the cloud content, propagating to AMF errors and finally to $NO_2$ retrieval errors. For this step, the $NO_2$ retrieval impacts were small but non-negligible for high $NO_2$ amounts; we estimated that a typical high $NO_2$ amount can cause a maximum retrieval error of $0.25 \times 10^{15}$ molecules/cm$^2$ for a PS of 5%. These simulation capabilities were designed to aid in the development of a GeoXO atmospheric composition instrument that will improve our ability to monitor and understand the Earth's atmosphere.

## 1 Introduction

NOAA's Geostationary Extended Observations (GeoXO) constellation will continue and expand on the capabilities of the current generation of geostationary satellite systems to support US weather, ocean, atmosphere, and climate operations. It is

planned to consist of a dedicated atmospheric composition instrument (ACX) to support air quality monitoring and forecast-
ing. The mission will build on knowledge obtained from low earth orbit (LEO) and geostationary (GEO) satellite air quality
monitoring instruments such as TROPOspheric Monitoring Instrument ( TROPOMI) (Veefkind et al. (2012)), OMI (Ozone
Monitoring Instrument) (Levelt et al. (2006, 2018)), Geostationary Environment Monitoring Spectrometer (GEMS) (Kim et al.
(2020)), and Sentinel 4 (Kolm et al. (2017)). Retrievals of trace gases like $NO_2$ derived from satellite platform observations
have been used to relate top-down emissions estimates, air quality monitoring and forecasting, pollution events, trends, and
health studies (Bovensmann et al. (2011); Levelt et al. (2018); Burrows et al. (1999); Bovensmann et al. (1999); Levelt et al.
(2006); Munro et al. (2016); Bak et al. (2017); Veefkind et al. (2012); Cooper et al. (2022); Hollingsworth et al. (2008)). The
World Health Organization has designated $NO_2$ as a pollutant, since it has detrimental effects on human health (WHO (2021);
Huangfu and Atkinson (2020)). It also impacts climate by contributing to the formation of aerosols in the upper troposphere that
reflect incoming solar radiation, and, thus, cool the planet (Shindell et al. (2009). Over non-polluted regions, the stratospheric
$NO_2$ participates in photochemical reactions that can affect the ozone layer (Crutzen (1979)).

In the near future, these phenomena will be monitored from geostationary (GEO) orbit over the greater North America as
part of the TEMPO (Troposphere Emission: Monitoring Pollution) mission (Zoogman et al. (2017)), at an increased temporal
frequency than available from its LEO counterparts. Like other atmospheric composition monitoring instruments, TEMPO is
and ACX will be a hyperspectral imager with fine spectral sampling and resolution from the ultraviolet to the near-infrared
allowing trace gas absorption features to be discriminated using the well-known differential optical absorption spectroscopy
(DOAS) technique. For total vertical $NO_2$ amount retrievals, the DOAS technique is applied around the 420 to 455 nm range
(Bucsela et al. (2006); Lamsal et al. (2021); Marchenko et al. (2015); Boersma et al. (2007); Richter and Burrows (2002); Valks
et al. (2011); Martin (2002)).

ACX is in its early stages of development with its initial performance requirements being formulated with respect to param-
eters like sampling and resolution to enable this DOAS approach. Other parameters such as pixel size, noise, and polarization
sensitivity (PS) are also being defined. These requirements may be updated as the instrument design choices are better un-
derstood. This study focuses on the requirements for instrument PS, which, for instance, may inform whether a polarization
scrambler is needed. Air quality monitoring instruments such as OMI and TROPOMI were designed with polarization scram-
blers to reduce their PS (Bézy et al. (2017); Voors et al. (2017)).

Without PS suppression, the polarization state of incoming radiation will impact the at-sensor radiance for satellites sensors
in both GEO (Pearlman et al. (2015)) and LEO, though these impacts have been more extensively analyzed for LEO satellites
(Meister and Franz (2011); Wu et al. (2017); Goldin et al. (2019)). GEO orbit presents unique challenges due to the highly
variable solar angles throughout the day. This results in a variation in the degree of linear polarization of the at-sensor radiance
throughout the day due to Rayleigh scattering in the Earth's atmosphere; for instance, light scattered in the normal direction
to the incident light generates highly polarized radiation but not in the forward or backward direction. If the instrument is
sensitive to light with a certain polarization, this variation in degree of linear polarization translates to a variation in measured
radiance throughout the day. Thus, limiting the PS of the satellite sensor can limit the radiometric uncertainty. These impacts

can be derived by employing radiative transfer simulations to predict the at-sensor polarization state or Stokes parameters ($S$) and applying the instrument polarization impacts via its Mueller matrix ($\mathbf{M}$).

$$\mathbf{S} = [\mathbf{S_0}\ \mathbf{S_1}\ \mathbf{S_2}\ \mathbf{S_3}]^{\mathbf{t}}$$
$$\mathbf{S}' = \mathbf{M}\,\mathbf{S} \tag{1}$$

The Stokes formulation expresses the polarization state consisting of its un-polarized (or randomly polarized) component, $S_0$; two terms describe its linear polarization state: the excess in horizontal linear polarization relative to the vertical direction, $S_1$, and excess in linear polarization at 45° relative to 135°, $S_2$; one term describing its circular polarization through its excess of right circular relative to left circular polarization, $S_3$. The Mueller matrix is a 4 x 4 matrix used to apply the optical effects of an element to generate an output Stokes vector. We model ACX as a Mueller matrix with a transmission of one and non-zero linear polarization extinction elements ($m_{01}, m_{02}, m_{10},$ and $m_{20}$). Since the system only detects total energy or radiance, not polarization state, only the first row is relevant. So the output term corresponding to the detected normalized Stokes parameter is:

$$S'_{ACX} = 1 + m_{01}S_1 + m_{02}S_2 \tag{2}$$

This detected radiance can differ from the true at-sensor radiance if ACX has linear PS, defined as $\sqrt{(m_{01}^2 + m_{02}^2)}$, which can propagate to higher level satellite products. For instance, the retrieval of surface reflectance can suffer large uncertainties, especially when the signal from the surface is small compared to the atmospheric component. In this work, we discuss our study of $NO_2$ retrievals, and investigate the parts of the process that may be affected. To our knowledge, $NO_2$ retrievals dependence on instrument PS have not yet been fully documented. We describe an initial study to show the ways that these retrievals can be impacted and make initial estimates of those impacts associated with the current PS requirements, $<5\%$ PS for wavelengths $<500$ nm.

Our $NO_2$ retrieval simulation approach discussed here follows a simplified version of the DOAS technique used for operational $NO_2$ retrievals and consists of two basic steps: One involves the DOAS spectral fitting step for the at-sensor radiance. This fit is normally used to retrieve the $NO_2$ slant column amount —the total number of molecules along the atmospheric photon path to the satellite sensor. The second step converts this slant column amount to the vertical column amount through the air mass factor (AMF), which depends on the geometrical path as well as the differences in scattering and absorption within the atmosphere between the slant and vertical paths. Our first approach for analyzing polarization effects deals with the DOAS spectral fitting step with clear sky scenes by simulating at-sensor Stokes parameters and applying an instrument model that includes a range of PS values in several orientations (defined by $m_{01}$ and $m_{02}$), as well as the instrument noise and spectral properties consistent with our current knowledge of ACX. The fits of these spectra are used to retrieve $NO_2$ vertical column amount directly, not slant column, in our case; since these are simulations with the vertical profiles used as inputs, we do not need to use the AMF for converting slant column to vertical column amount. The second approach deals exclusively with the AMF derivation step. For this analysis, the AMF, required for operational retrievals, is affected by instrument PS when

considering the potential for partially cloudy scenes. Retrievals in such situations are commonly performed for atmospheric monitoring instruments, since their large instantaneous fields of view make completely clear scenes rare. We will discuss the formalism in detail for both approaches in the methods section. With these two approaches, referred to as the method for "clear scenes" and "partially cloudy scenes", we demonstrate the capability to investigate PS requirements.

## 2 Methods

As mentioned, the approach for clear scenes exploits the spectral features in the radiance spectra to retrieve the total vertical amount of $NO_2$, and the approach for cloudy scenes relies on the AMF calculation.

### 2.1 Clear scenes

The overall method for clear scenes is illustrated in Fig. 1. In this process, simulated radiance spectra are propagated through an instrument model and the total vertical column $NO_2$ is retrieved using a look-up table (LUT) approach with the aid of a constrained energy minimization algorithm (CEM) algorithm (Farrand (1997)). Further details are discussed below.

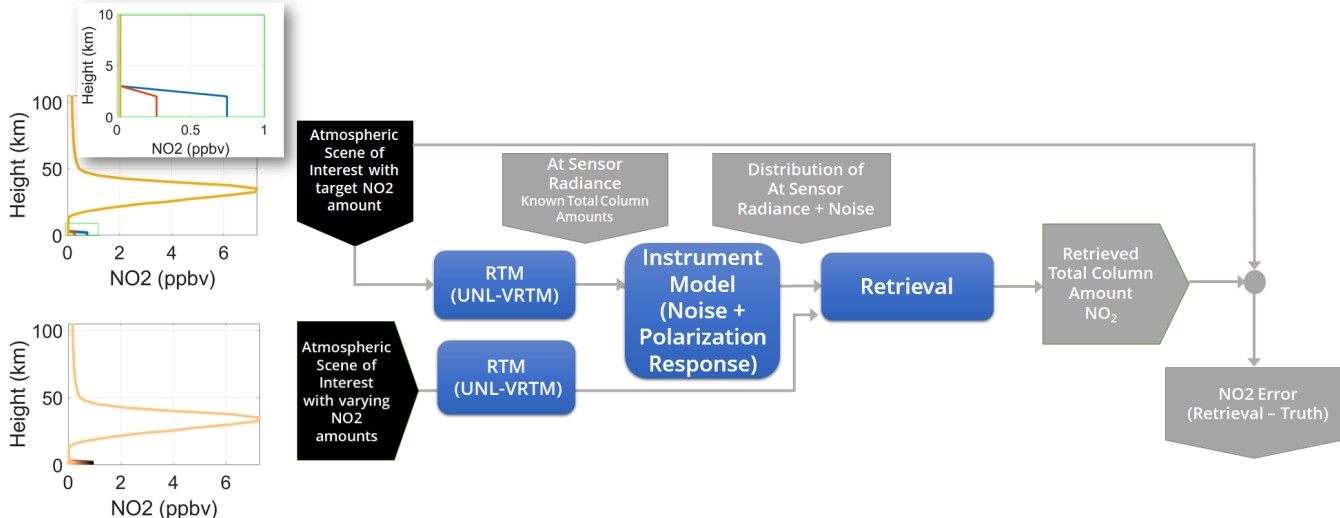

**Figure 1.** Simulation method for retrieving $NO_2$ in clear scenes: The scenes of interest consist of selected custom $NO_2$ profiles to represent low, medium, and high $NO_2$ cases shown at the upper left (including a zoomed-in view) corresponding to total vertical $NO_2$ amounts of $4.60, 5.93,$ and $8.44 \times 10^{15}$ molecules/cm$^2$, respectively. The lower left profiles contain all profiles used in the retrieval process. The profiles are used in the radiative transfer model (RTM) called the Unified Linearized Vector Radiative Transfer Model (UNL-VRTM) to generate at-sensor radiance.

### 2.1.1 Radiative Transfer Modeling

The at-sensor radiances from clear scenes are simulated using a vector radiative transfer code, the Unified Linearized Vector Radiative Transfer Model, UNL-VRTM, which integrates the linearized vector radiative transfer (VLIDORT) into a broader framework (Xu and Wang (2019)). The code can generate Stokes vectors from any scene defined by its view and solar geometry, surface reflectance, wavelength range, and atmospheric composition. Note that rotational Raman scattering is not included in the model. The ACX was assumed to be at 105° West longitude viewing several locations across the continental US (CONUS). The time of day was chosen to generate solar zenith angles of 60 to 70°, where PS is expected to be highest but still within the range where $NO_2$ retrievals are typically performed. The US Standard Model default profiles were used for 21 trace gases for all scenes (excluding $NO_2$). The default $NO_2$ profiles were modified by injecting a known amount uniformly into the troposphere below 2 km (Fig. 1). Three basic surface spectra generated from spectral libraries were used. The water spectrum used is associated with an open ocean case (Kokaly et al. (2017)); the vegetation is a combination of trees (30 %), grass (30 %), shrubs (30 %), non-photosynthetic material (5 %), and soil (5 %), and the urban case is a combination of roof (50 %), concrete (20 %), road (20 %), and vegetation (10 %) (Meerdink et al. (2019); Baldridge et al. (2009)) as depicted in Fig. 2. Their associated background aerosol content was included in the boundary layer up to 2 km with a uniform vertical distribution. The rural and urban scenes use a bi-modal aerosol distribution as shown in Table 1, where the loading and size distribution values for each mode are given for these scenes. The aerosol parameters including the complex indices of refraction per wavelength were taken from Shettle et al. (1979) (with mean values listed in the Table) and aerosol optical depth (AOD) values from the climatology reported in Yan et al. (2021).

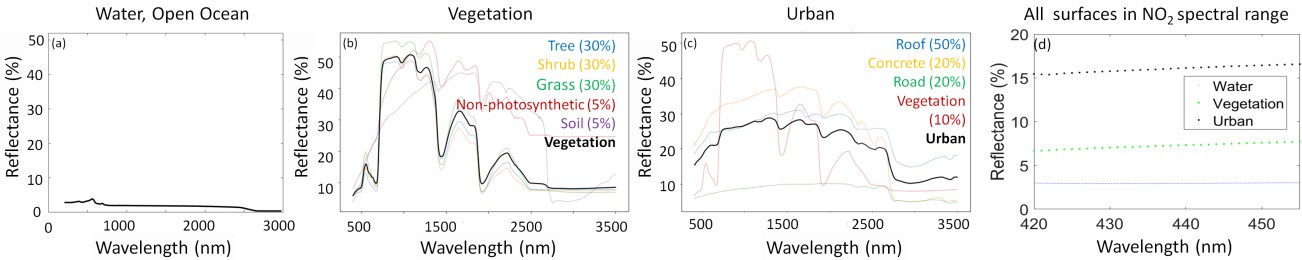

**Figure 2.** Basic surfaces reflectance spectra used in radiative transfer simulations: (a) water, (b) vegetation and (c) urban. (d) Spectra in the $NO_2$ retrieval spectral range.

We ran radiative transfer simulations for several US locations, with the three scene types, with varying amounts of tropospheric $NO_2$. This produced a look up table (LUT) of scene type, $NO_2$ vertical amount, and at-sensor radiance spectra. This LUT was used in the retrieval discussed below.

**Table 1.** UNL-VRTM Parameters

| Scene | Reflectance Spectrum | Aerosol | | | | | | |
|-------|----------------------|---------|---|---|---|---|---|---|
| | | Type(s) | Index of refraction | | AOD [1] | Size distribution | | |
| | | | Real | Imaginary | | $\bar{r}$ [$\mu$m] [2] | $\sigma_r$ [$\mu$m] [3] | |
| Water | Open ocean | Sea Salt | 1.50 | 0.0 | 0.08 | 0.3 | 0.4 | |
| Rural | Trees | Water soluble | 1.53 | 0.0050 | 0.13 | 0.03 | 0.35 | |
| | Shrubs | Dust | 1.53 | 0.0049 | 0.42 | 0.5 | 0.4 | |
| | Grass | | | | | | | |
| | Non-photosynthetic | | | | | | | |
| | Soil | | | | | | | |
| Urban | Roof | Water soluble | 1.53 | 0.0050 | 0.03 | 0.03 | 0.35 | |
| | Concrete | Soot | 1.75 | 0.456 | 0.5 | 0.5 | 0.40 | |
| | Road | | | | | | | |
| | Vegetation | | | | | | | |

[1] aerosol optical depth

[2] radius mean

[3] radius standard deviation

## 2.1.2 Instrument model and NO$_2$ retrievals

The reference radiance spectra corresponding to the NO$_2$ reference amounts over water, rural and urban scenes were modified by applying the instrument model (for several US locations). The instrument response model was based on the TEMPO design, which consists of of a reflective f/3 Schmidt-form telescope and a spectrometer assembly that utilizes a diffraction grating

to form an image on CCD detector arrays (Zoogman et al. (2017)). The simulated radiance was modified by this instrument response model, which sampled the radiance at 0.2 nm wavelength steps with a resolution of 0.6 nm, and applied a PS response. The PS response model was not specific to TEMPO as our goal was to understand the range of impacts associated with the ACX polarization requirements. The noise was also applied as defined by the ACX signal-to-noise (SNR) specification. Our instrument parameters from TEMPO were modified by assuming a sampling strategy or integration time modification that

brought the noise in line with that specified by ACX. Table 3 shows the parameters included in this model.

**Table 2.** ACX instrument response model parameters

| Parameter | Description |
|-----------|-------------|
| $L(\lambda)$ | Spectral radiance at instrument resolution |
| $A_{det}$ | Detector area |
| $\Omega = \pi/4(f\#)^2$ | Solid angle of acceptance |
| $\Delta t$ | Integration time |
| $\lambda$ | Wavelength |
| $\Delta\lambda$ | Spectral interval per pixel |
| $\tau(\lambda)$ | Optical system transmittance combined with grating efficiency |
| $\eta(\lambda)$ | Detector quantum efficiency |
| $N$ | Bit depth |
| $n_{\text{read}}$ | Read noise |
| $I_{\text{dark}}$ | Dark current |

The noise was applied by generating 1000 spectra with different amounts of noise following a Gaussian distribution that are added to the at-sensor radiance (after being modified by the polarization response). All spectra were normalized by subtracting a second order polynomial fit to remove the sensitivity to absolute radiance as is done in the DOAS retrieval technique. The $NO_2$ vertical amount was retrieved using the look-up-table and the CEM algorithm:

$$
\text{CEM} = \frac{(t-m)^T C^{-1} (x-m)}{(t-m)^T C^{-1} (t-m)}, \tag{3}
$$

where $C^{-1}$ and $m$ are the inverse covariance matrix and mean over the noise spectra, respectively. The CEM was calculated for all (target) spectra in the LUT, $t$, with the noise spectra, $C^{-1}$ and $m$. The spectrum, $x$ that generated a CEM value closest to one was chosen, and its associated $NO_2$ vertical amount was retrieved.

## 2.2 Partially cloudy scenes

The process for "partially cloudy scenes" involves an AMF derivation process that includes the consideration of subpixel-scale clouds. The typical instantaneous field of view for an atmospheric composition instruments means that most scenes contain some clouds. Operational trace gas retrievals are routinely done in partially cloudy scenes, so we derive PS impacts for such scenes primarily through their impact on the AMF.

### 2.2.1 Theoretical background

This approach assumes a simple cloudy scene model where each scene is assumed to be a combination of a fully cloud covered subpixel and a clear sky subpixel weighted with an effective cloud fraction, $f$, consistent with previous approaches (Stammes et al. (2008)):

$$
L_{obs} = L_{clr}(1-f) + L_{cld}f, \tag{4}
$$

where $L_{obs}$ is the observed radiance, $L_{clr}$ is the calculated radiance in a clear sky, and $L_{cld}$ is the cloudy radiance. To produce observed amounts of Rayleigh scattering and absorption, it was found that for this equation to work across most conditions, we model $L_{cld}$ as a Lambertian surface (opaque) with surface reflectivity 0.80 at the effective cloud pressure, assumed here to be equivalent to a cloud at 2 km. Aerosols are not considered for the cloudy scenes, since they would have a negligible impact; the clouds would lie above the tropospheric NO2 and aerosol layer. This simple model has been demonstrated to represent the complex radiative transfer in clouds accurately (Stammes et al. (2008); Joiner (2004); Vasilkov et al. (2008)). So, we typically derive $f$ at a wavelength with little absorption and use a surface climatology for $L_{clr}$. Then, we simply invert the above equation to give:

$$
f = \frac{L_{obs} - L_{clr}}{L_{cld} - L_{clr}}. \tag{5}
$$

For the trace gas retrievals, another quantity defines the fraction of scene radiance from the cloud versus the clear parts of the scene called the cloud radiance fraction, $f_r$, which has wavelength dependence:

$$f_r = f \frac{L_{cld}}{L_{obs}}. \tag{6}$$

A cloudy air mass factor (AMF) is computed along with the clear sky AMF. The total AMF is then computed with the clear and cloudy AMFs weighted by the cloud radiance fraction

$$\text{AMF}_{total} = \text{AMF}_{clr}(1 - f_r) + \text{AMF}_{cld}f_r. \tag{7}$$

To compute the error in the $\text{NO}_2$ vertical column due to an error in $f$, we started with the calculation of the error in $f$ due to an error from PS:

$$\frac{df}{d\epsilon_{PS}} = \frac{dL_{obs}}{d\epsilon_{PS}} \frac{1}{(L_{cld} - L_{clr})}, \tag{8}$$

and this would then propagate into the error in $\text{NO}_2$ vertical column density ($NO_{2,VCD}$) through Equations 6,7 above along with:

$$\text{NO}_{2,\text{VCD}} = \frac{\text{NO}_{2,\text{SCD}}}{\text{AMF}_{total}}. \tag{9}$$

This process is shown graphically in Fig 3, where a clear and cloudy version of a scene are simulated. The clear version is propagated through the instrument polarization response model, and, using the radiance generated from the cloudy scene, the impacts are propagated through the cloud fraction, cloud radiance fraction, AMF, and finally the $\text{NO}_2$ amount. Following the process by Kuhlmann et al. (2015), the AMFs for each atmospheric layer (also called box AMFs) were computed using a pre-calculated LUT with input parameters of altitude, $z$, solar zenith angle, view zenith angle, relative azimuth angle, surface reflectance, and surface altitude. The total AMF was calculated by linearly interpolating over all variables for each altitude and summing over all layers to the top of atmosphere (TOA), where each layer $dz$ has a vertical column amount $V_{NO_2}$:

$$\text{AMF}_{clr/cld} = \frac{\int_0^{TOA} \alpha \cdot \text{AMF(z)} \cdot V_{NO_2} dz}{\int_0^{TOA} V_{NO_2} dz}, \tag{10}$$

where the integration assumes an exponential dependence within each layer. A correction term, $\alpha$, is normally included in the AMF calculation to account for the temperature dependence of the $\text{NO}_2$ cross sections, though was neglected here by setting it to one. The $\text{NO}_2$ error derived through the conversion of slant to vertical amount is then computed. This error can be considered as the effect of a change in detected radiance due to PS, which, in turn, leads to an error in the interpretation of the amount of clouds in the scene. This leads to an impact on the $\text{NO}_2$ retrieval over the total vertical column. Note that assuming a constant PS over the wavelength range, this error will also change negligibly as a function of wavelength. We perform this

analysis at one wavelength (425.8 nm) in this study. By differentiating Equation 9, the $NO_2$ error in the total vertical column amount ($\partial(NO_{2,total})$) is then calculated in terms of the total vertical $NO_2$ amount ($V_{NO_2,total}$), the AMF, and the AMF error ($\partial AMF_{total}$) as:

$$\partial(NO_{2,total}) = \frac{-V_{NO_2,total}}{AMF_{total}} \cdot \partial(AMF_{total}). \tag{11}$$

### 2.2.2  Radiative transfer modeling

We conducted the radiative transfer simulations as summarized in Table 3. Simulation A will be shown to define an upper bound for the retrieval error with a PS of 5% by using a $NO_2$ profile (similar to those defined in the clear scene simulations) with a large $NO_2$ amount, the lowest reflectance scene, and high constant solar zenith angle over all of CONUS over a one degree latitude/longitude grid. Simulation B quantifies the retrieval impact of scene type —water, rural, and urban scene —over CONUS for a constant reference $NO_2$ profile. The scene types are the same as defined in Table 1 and are assigned to all pixels in CONUS for each run. Simulation C explores the retrieval impacts on the solar zenith angle and $NO_2$ amount for selected US locations. The PS is also varied over a wider range of values. Finally, Simulation D uses $NO_2$ profiles from the Goddard Earth Observing System Model, Version 5 (GEOS-5) (Molod et al. (2012)) on a particular time and day with a fixed scene type over the CONUS grid. Simulations A-C give a contrived version that is useful for bounding the impacts of instrument PS and isolating impacts of different variables. Simulation D represents cases with more realistic nominal parameters. Note that we also used the cloud fraction from the GEOS-5 model for deriving the simulated radiance prior to applying the polarization response model. This deviates from the illustration in Fig. 1 (top left), where instead of a clear scene, a mixture of cloudy and clear scene according to the GEOS-5 cloud fraction value is used, thereby accounting for the radiance polarization state of both clear and cloudy scenes in generating the $NO_2$ retrieval errors. A single day was chosen to demonstrate this approach, July 15, 2007 on two selected times 16 UTC and 20 UTC, so that the impacts of extreme solar zenith angles (corresponding to high degree of linear polarization) could be seen for both the eastern and western US regions.

**Table 3.** ACX Radiative Transfer Simulation for Cloudy Scenes

| Simulation | NO$_2$ amount | Solar zenith angle/Time | Scene | Polarization Sensitivity (PS) | Orientation | Locations |
|---|---|---|---|---|---|---|
| A | $20 \times 10^{15}$ molecules/cm$^2$ | 70° | Water | 5 % | Vertical | CONUS[1] |
| B | $8.4 \times 10^{15}$ molecules/cm$^2$ | 70° | Water | 5 % | Vertical | CONUS |
|  |  |  | Vegetation |  | 45° |  |
|  |  |  | Urban |  |  |  |
| C | $20 \times 10^{15}$ molecules/cm$^2$ | 70° | Water | Variable | Vertical | Select locations |
|  | $8.4 \times 10^{15}$ molecules/cm$^2$ | 30° |  |  | Horizontal |  |
|  | $5.9 \times 10^{15}$ molecules/cm$^2$ |  |  |  |  |  |
| D | GEOS-5[2] profiles | 16 UTC | Water | 5% | Vertical | CONUS |
|  |  | 20 UTC |  |  |  |  |

1 Continental United States, 2 Goddard Earth Observing System Model, Version 5

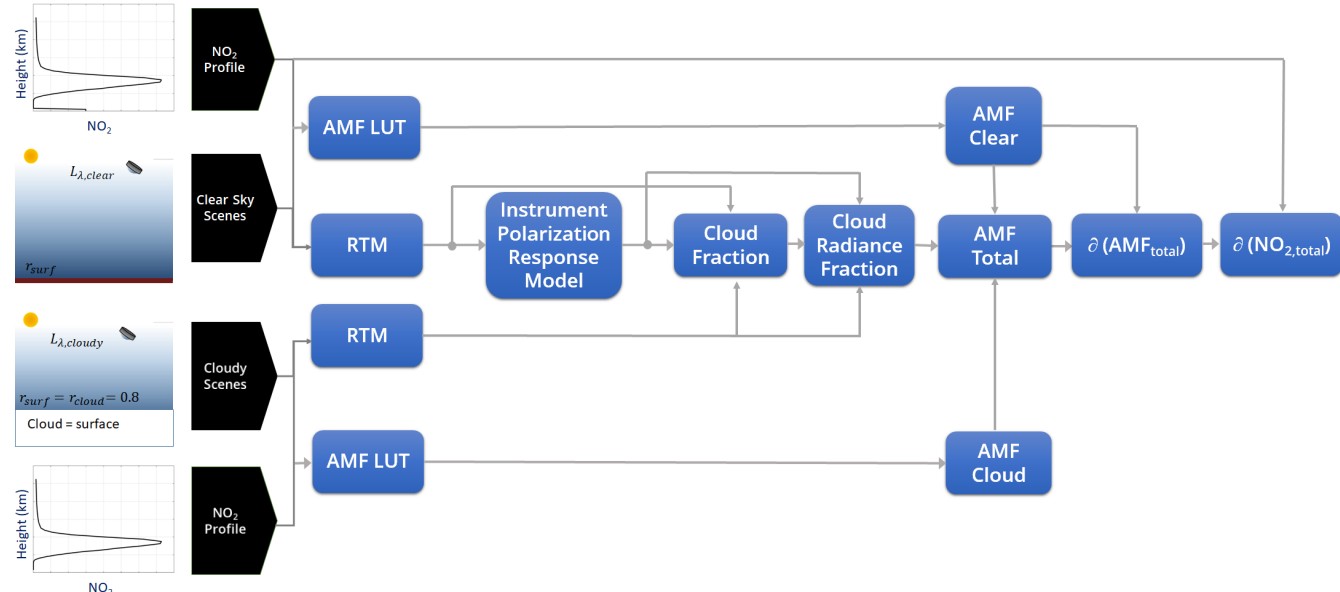

**Figure 3.** Simulation method for deriving $NO_2$ errors by interpreting a clear scene as a partially cloudy scene due to instrument PS: Through radiative transfer modeling (RTM) and air mass factor (AMF) calculations via a look-up table (LUT) of clear and cloudy scenes, and applying the instrument polarization response model to a clear scene, the $NO_2$ error is determined by propagating through the variables shown to errors in AMF ($\partial(AMF_{total})$) and total vertical $NO_2$ amount ($\partial(NO_{2,total})$)

## 3 Results

### 3.1 Clear scenes

As part of the method for clear scenes, the ACX instrument model was applied to the at-sensor radiance including sampling with a Gaussian slit function at the interval and resolution of 0.2 and 0.6 nm, respectively, and its noise as depicted in Fig. 4. The differences between the normalized solar irradiance (multiplied by a factor of 5 for visibility) and radiance spectra shows the atmospheric contribution and the effects of this resampling. The 1000 radiance spectra shown cannot be discerned clearly given the high SNR (explicitly shown by the blue line). The noise was applied after modifying with the PS response. The PS model parameters applied via Equation 2 using $m_{01} = \pm PS$ and $m_{02} = 0$, so that the PS was applied in the vertical or horizontal orientation. These orientations were chosen for most simulations for simplicity but other orientations will be discussed in the cloudy scene analysis section.

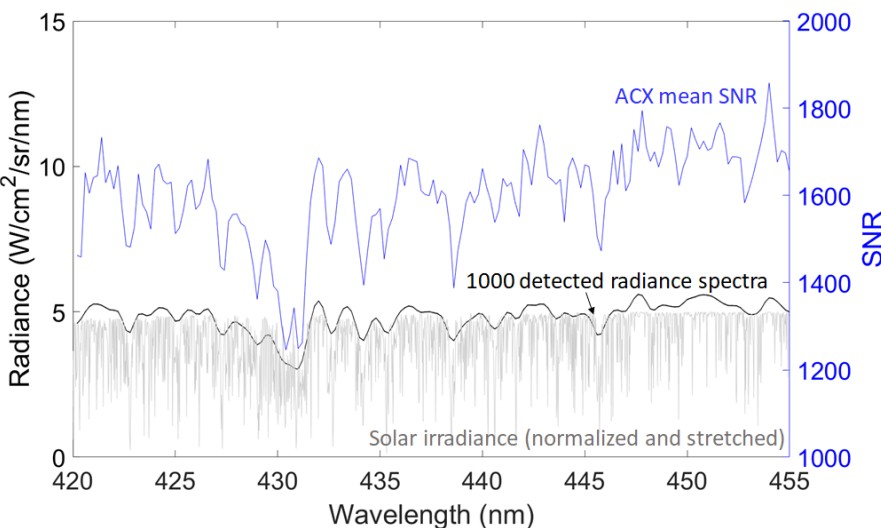

**Figure 4.** Example of at-sensor radiance spectra simulated with an applied instrument model including resampling effects and added noise set by ACX instrument parameters. 1000 spectra are plotted (black lines), which appears as a slightly thicker line than the mean SNR (blue line and right axis). The normalized solar irradiance multiplied by a factor of 5 is shown for comparison to the resampled spectra.

The retrieval process effectively matches the spectral shape of the simulated detected spectra —affected by spectral sampling, noise, and PS —to the most similar spectra in the LUT that contains a large range of tropospheric $NO_2$ amounts for the three surfaces. Figure 5(a) shows an example of a the adjusted sample spectrum with the the spectra in the LUT. Note that all spectra were adjusted using quadratic fits in the spectral fitting process. The CEM algorithm finds the spectrum from the spectra that is most similar. Figure 5(b) shows a summary of the $NO_2$ retrieval errors, average biases and standard deviations as a function of

PS for several scene types for a particular location (Norman, Oklahoma). The errors are driven by a combination of the SNR, view/solar geometry, surface reflectance spectrum, and aerosol model and are similar for all scene types. The flat dependence indicates that the PS does not affect the retrieval error in the DOAS spectral fitting retrieval step. The reason is that the PS is a smooth function of wavelength, and the radiometric error introduced are compensated through the spectral fitting process. These results were similar for all locations (not shown). We note that other retrieval techniques that do not use a polynomial

correction term in the spectral fitting approach may exhibit larger PS impacts.

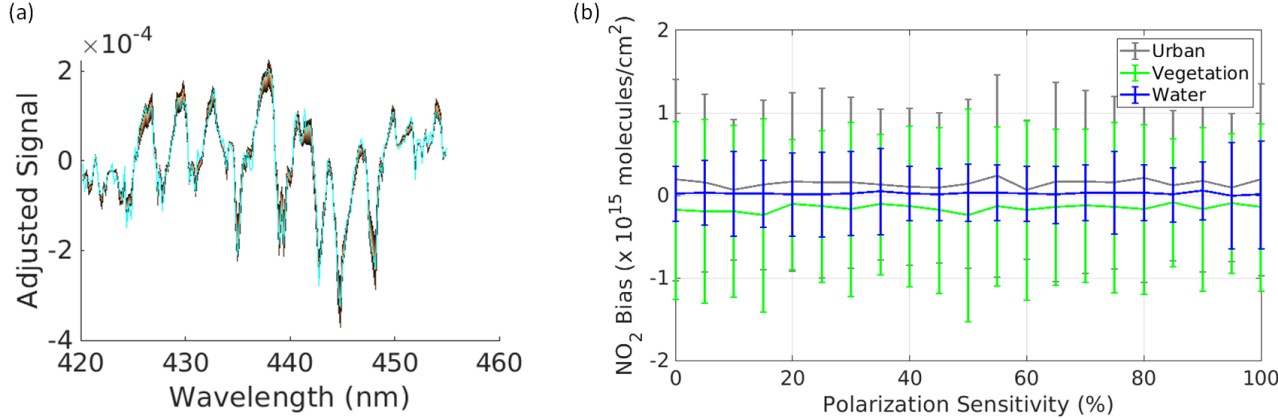

**Figure 5.** Clear-sky scene retrieval results: (a) An example of an adjusted ACX simulated spectrum (cyan) with all spectra from the look-up-table (LUT) with varying amounts of tropospheric $NO_2$ (b) The average error (or bias) and standard deviation for 1000 total vertical $NO_2$ retrievals of the "high" amount ($8.44 \times 10^{15}$ molecules/cm$^2$) for the three scene types (water, vegetation, and urban) at Norman, Oklahoma, assuming a vertical PS orientation.

### 3.2 Partially cloudy scenes

In contrast to the previous results, the AMF-related processing step showed more significant polarization impacts, where an error is induced when a clear scene scene is interpreted as a partially cloudy scene due to the instrument response model that includes PS (but not noise). Figure 6 shows the results as they are propagated through each step in the process (Fig. 3) for an
230 example with an extremely high total vertical $NO_2$ amount, $20 \times 10^{15}$ molecules/cm$^2$, over all of CONUS (Table 3, Simulation A). The simulation ran using $70°$ solar zenith angle and water scene for all pixels and an instrument PS of $5\%$, $m_{01} = -0.05$, vertical orientation and $m_{02} = 0.05$, $45°$ orientation, and an initial cloud fraction of zero. The Stokes parameter, $S_1$ is relevant for vertical (or horizontal) polarization and $S_2$ is relevant for $45°$ (or $135°$) polarization. The correlation between the relevant Stokes parameters, retrieved cloud fraction, and $NO_2$ error are particularly apparent. This example shows that the PS orientation
can generate vastly different spatial dependence in $NO_2$ retrieval errors. The maximum $NO_2$ error of $1.4 \times 10^{15}$ molecules/cm$^2$ is above the specified TEMPO $NO_2$ precision (Zoogman et al. (2017)). Note that this is likely an upper bound, since $NO_2$ amounts like these are mostly found in industrialized areas in other regions of the world.

Similar simulations for more realistic $NO_2$ amounts using constant profiles across CONUS show how these retrieval errors change as a function of surface type (Table 3, Simulation B). Figure 7 shows a lower, more realistic, $NO_2$ amounts of $8.4 \times$
$10^{15}$ molecules/cm$^2$ corresponding to the "high" $NO_2$ case shown in Fig. 1. The results are shown for the three different scene types applied uniformly across all of CONUS. The other parameters are the same as the previous higher $NO_2$ case. The $NO_2$ error increases as the surface reflectance decreases. All cases show the same spatial pattern over CONUS as in the previous case. The maximum $NO_2$ error is $0.25 \times 10^{15}$ molecules/cm$^2$.

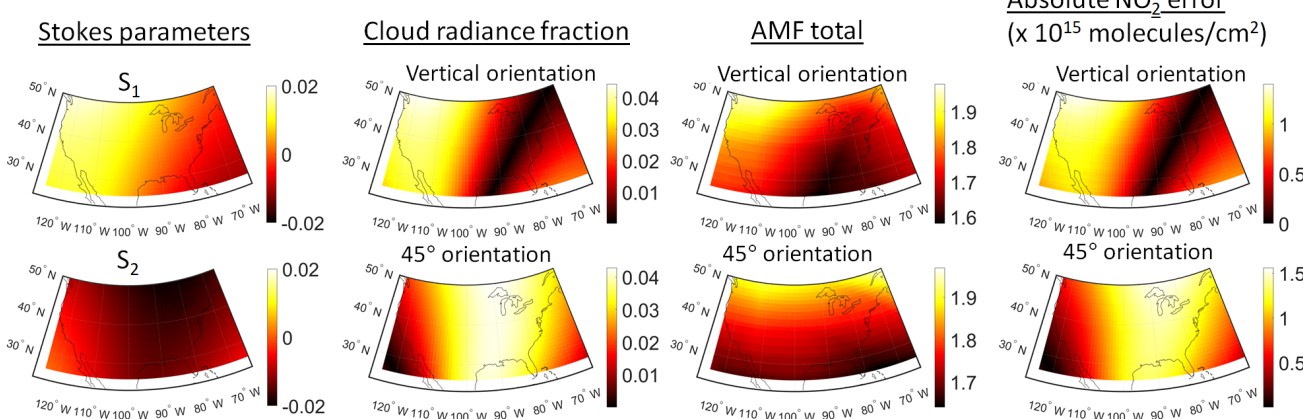

**Figure 6.** Derived parameters for $NO_2$ amount of $20 \times 10^{15}$ molecules/cm$^2$ , water scenes, and 5% PS in a vertical and 45° orientation. (See Table 3, Simulation A for more details).

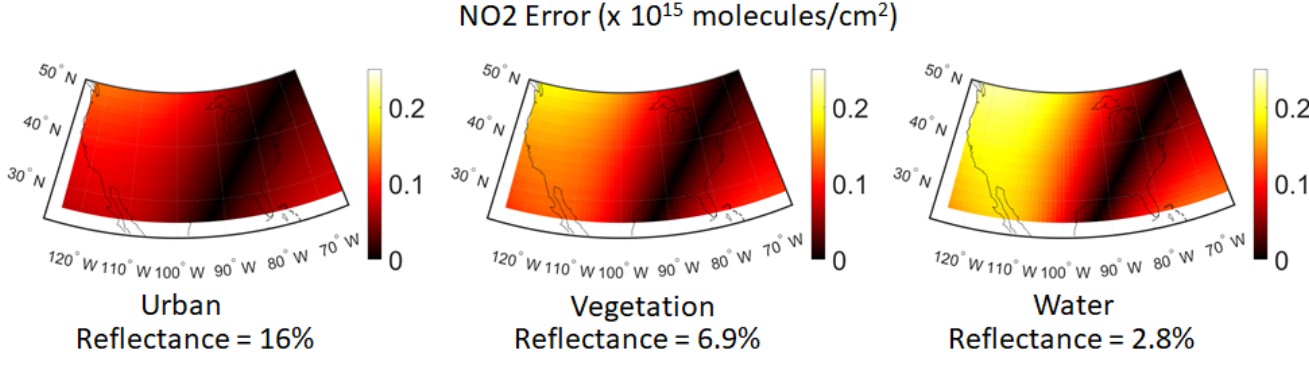

**Figure 7.** $NO_2$ errors assuming different scene types across CONUS for 5% PS in a vertical orientation and constant $NO_2$ profiles with 8.4 $\times 10^{15}$ molecules/cm$^2$. (See Table 3, Simulation B for more details).

Fig. 8 shows the results repeating similar simulations with different $NO_2$ amounts and times of day for select US locations
and their (non-linear) dependence on PS (Table 3, Simulation C). The figure shows that $NO_2$ errors derived decrease as the $NO_2$ amounts decrease using three different total vertical amounts: 5.9, 8.4, and $20 \times 10^{15}$ molecules/cm$^2$ as a function of PS and two different orientations. The dependence on $NO_2$ amount is non-linear; for instance, at 5 % PS for the Seattle, evening case, the retrieval errors for increasing amounts are 0.22 %, 2.6 %, and 6.6 %. The time of day dependence is illustrated by the edge of the shading: the darker shading shows the retrieved $NO_2$ amount with a solar zenith angle of 30° and the edge
of the lighter shading shows the amount with an angle of 70°. The shading is meant to emphasize the difference between the reference and retrieved amount. The horizontal orientation results are similar to those for the vertical orientation. As evident in

the previous results, the largest $NO_2$ errors occur in the western regions (Seattle, San Diego) for these orientations. The lower solar zenith angle corresponds to lower linear degree of linear polarization, accounting for the lower $NO_2$ errors.

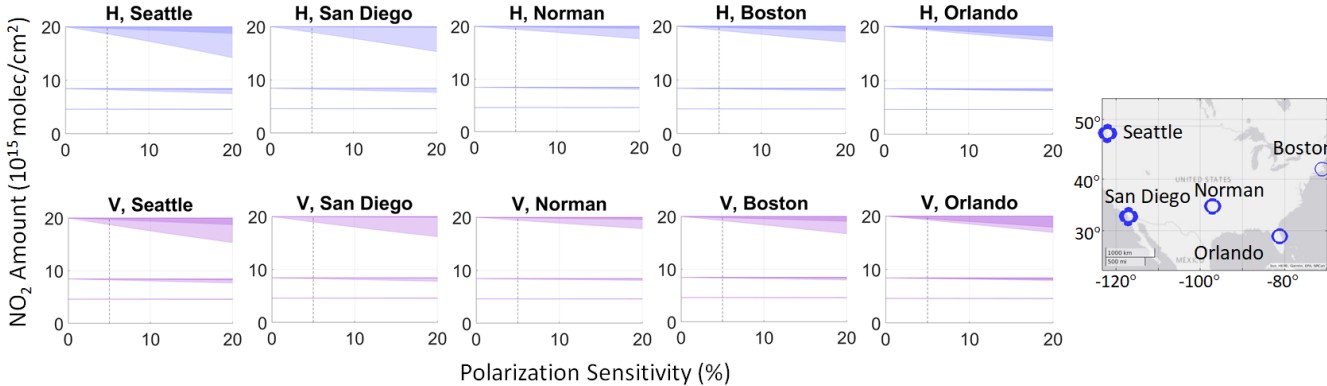

**Figure 8.** (Left) The retrieved amount is shown as a function of polarization sensitivity (PS) for two different orientations (H - horizontal, V - vertical) for selected US locations and $NO_2$ total vertical amounts: 5.0, 8.6, and 20 $\times 10^{15}$ molecules/cm$^2$. The edge of the darker (lighter) shading shows the retrieved $NO_2$ amount with a solar zenith angle of 30° (70°). The vertical dotted line shows the current PS requirement for reference. (Right) The locations are shown on the map with thicker circles representing higher $NO_2$ errors. (See Table 3, Simulation C for more details)

In contrast to the previous results with constant profiles across CONUS, Fig. 9 shows the results using GEOS-5 profiles, which appear qualitatively consistent with the results using the artificial profiles used above (Table 3, Simulation D). The $NO_2$ amounts for this day varied between 2.5 to 6.5 $\times 10^{15}$ molecules/cm$^2$ are displayed . The figure shows the polarization impacts with 5% PS in the vertical orientation. The impacts are more apparent as the solar zenith angle increases and resemble the previous results in Fig. 7, where the solar zenith angle is fixed at 70 °. For instance, the $NO_2$ errors are larger at 20 UTC in the eastern regions where the solar angles are relatively large, and the $NO_2$ errors are larger in the western regions at UTC 16, where the solar zenith angles are larger. The higher cloud fraction decrease the retrieval errors, which can be seen in the western regions at 16 UTC; although the southwest and southeast have similar solar zenith angles, the southwest has lower retrieval errors due to the increased cloud fraction. As a result of the cloud fraction and lower $NO_2$ amount, the maximum $NO_2$ errors found were 0.03 $\times 10^{15}$ molecules/cm$^2$ for this day —a negligible value when compared to the TEMPO precision requirement.

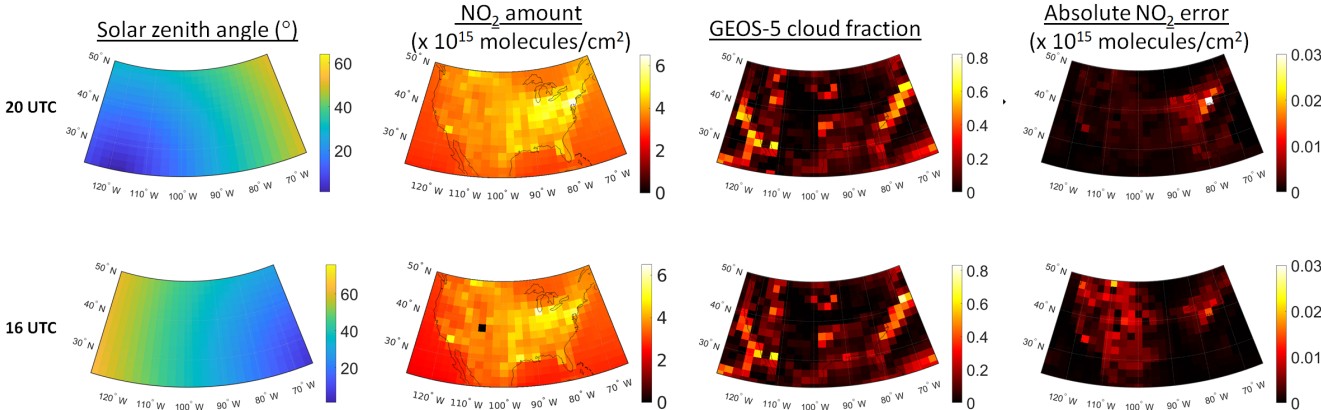

**Figure 9.** Solar zenith angles, total $NO_2$ column amount, GOES-5 cloud fraction, and resulting $NO_2$ errors at 20 UTC (top) and 16 UTC (bottom). GEOS-5 $NO_2$ profiles were used assuming 5% PS with vertical orientation, all water scenes, and clouds at a 2 km altitude. (See Table 3, Simulation D for more details)

## 4  Summary and conclusions

We demonstrated a simulation and modeling capability to assess polarization effects for ACX predicted performance studies. Our results show that the DOAS spectral fitting step mitigates PS effects in the $NO_2$ retrieval process. The AMF calculation step, however, can cause retrieval errors from instrument PS when considering partially cloudy scenes. The PS magnitude and orientation (Mueller matrix elements) impacts can cause different $NO_2$ retrieval errors depending on location, time of day, cloud fraction, and $NO_2$ amount. For a PS of 5 % with vertical orientation, the maximum $NO_2$ retrievals errors were $0.25 \times 10^{15}$molecules/cm$^2$ for high pollution cases. In extreme cases, if $NO_2$ pollution significantly increases to levels on the order of the world's most polluted regions, these errors can reach $1.4 \times 10^{15}$molecules/cm$^2$. A more typical maximum error found through analyzing the GEOS-5 profiles was $0.03 \times 10^{15}$molecules/cm$^2$. This study shows that in most cases, the 5% PS requirement introduces retrieval uncertainties significantly lower than the TEMPO precision requirement except in the most extreme cases. Note that these estimates assume a particular set of instrument Mueller matrix elements. We emphasized a vertical orientation based on an assumed vertical grating orientation where its polarization axis would likely be in this direction. In this configuration, the instrument effectively sweeps wavelengths over locations in the west-east direction. The Mueller matrix will be updated with the appropriate values as the instrument design matures to refine the estimates of $NO_2$ retrieval impacts. Our simplified retrieval approach may have neglected factors used in operational retrievals that could be affected by instrument PS and contribute to additional retrieval errors related to estimates of aerosols, surface reflectance, and cloud parameters. Rotational Raman scattering, which has been used in cloud height retrievals (e.g., Vasilkov et al. (2008)), for instance, can be particularly sensitive to polarization. Other approaches for cloud height retrievals such as oxygen dimer absorption (Acarreta et al. (2004)) should be much less sensitive. We do not account for the PS to cloud height retrievals. The PS to cloud optical thickness is implicitly accounted for within the effective cloud fraction estimation. In addition, the limited

set of surface reflectance types that were used and the directional and polarization surface effects that were neglected, can be included in future work to improve the accuracy of the results. This capability can be utilized to support the development of ACX to continue and build on the legacy of atmospheric composition measurements to forecast and monitor air quality.

*Author contributions.* Conceptualization, J.J., J.M., A.P., F.P.; methodology, J.J., A.P., M.C., B.E., L.L.; formal analysis, A.P., M.C.; resources, J.J., J.M. ; writing—original draft preparation, A.P.; writing—review and editing, A.P., J.J., M.C., F.P., L.L. ; software, B.E., L.L.; supervision, J.J., J.M.; project administration, J.J., J.M. All authors have read and agreed to the published version of the manuscript.

*Competing interests.* We declare that the authors have no conflicts of interest.

*Acknowledgements.* Xiong Lu and Kelly Chance assisted with ACX instrument model parameterization. Xiaoguang Xu and Jun Wang supplied and assisted with the UNL-VRTM code.

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
