# Peer review of "Polarization Performance Simulation for the GeoXO Atmospheric Composition Instrument: NO2 Retrieval Impacts"

_EGUsphere, 2022_

## Referee Comment (RC1)

**Comment on egusphere-2022-207 by Anonymous Referee #1**

Referee comment on "Polarization Performance Simulation for the GeoXO Atmospheric Composition Instrument: NO2 Retrieval Impacts" by Aaron Pearlman et al., EGUsphere, https://doi.org/10.5194/egusphere-2022-207, 2022.

In this manuscript, the authors report on a study of the impact on $NO_2$ retrieval of instrument polarization of the Atmospheric Composition Instrument (ACX) planned for NOAA's Geostationary Extended Observations (GeoXO) constellation specifically by using currently known instrument specifications. This investigation is very relevant in case ACX will not be equipped with a polarization scrambler. The paper is written well and the conclusions that can be drawn from the stude are written clear and concise, but the method description could do with some more details.

**Major comments**

The description of the instrument (response) model in Sect. 2.1.2 gives very little details and there is no reference to a more detailed description. How does polarization influence the measurement? What kind of spectrally-dependent polarization features may be expected? What knowledge is available from GEMS and TEMPO, instruments that do not have a polarization scrambler? Could a figure be provided showing polarization sensitivity (PS) vs. wavelength?

**Minor comments**

- Page 1, line 5: Given the formulation of names and acronyms of the other instruments, it is more logical to write "... and OMI (Ozone Monitoring Instument), ..."

- Page 2, line 26: Given the formulation of names and acronyms of the other instruments, it is more logical to write "... such as TROPOspheric Monitoring Instrument (TROPOMI), ..."

- Page 2, line 27–28: The "GEMS" discussed by Hollingsworth is an entirely different GEMS than the Korean geostationary spectrometer. Please add an appropriate reference for GEMS. The paper of Hollingsworth is more appropriate for line 30.

- Page 2, line 35: The paper by van Geffen et al. (2020) describes the TROPOMI NO2 retrieval and is more appropriate in line 26.

- Page 2, line 51: Since both GEO and LEO have been spelled out in line 25, the line can read "... in both GEO (...) and LEO, though ..."

- Page 3, line 62: For clarity please write "... its circular polarization through ..."

- Page 4, Sect. 2.1.1: The light scattered by the surface may also become partly polarized. An example is the scattering by a water surface (e.g. principle behind Polaroid sun glasses). Is this taken into account, and could this affect the analysis?

- Page 5, line 110: A space is missing in "shrubs(30%)"

- Page 5, Table 1: Please explain the parameters AOD and sigma. Are the aerosols a mix of scattering and absorbing aerosol types?

- Page 6, line 132: The second occurrence of "all" at the start of the line needs to be removed

- Page 7, line 145: Surely the cloudy radiance $L_{cld}$ is meant here.
  Are these radiances polarized, or unpolarized? Would that influence the result?

- Page 7, line 150: A comma is missing in "... cloud radiance fraction, $f_r$, ..."

- Page 8, lines 171–172: The "temperature correction" appears out of nowhere and it is not clear what it refers to. Is it related to the temperature dependence of the $NO_2$ cross sections, which is compensated by a temperature correction term in the AMF calculation in the $NO_2$ retrievals of e.g. OMI and TROPOMI?

- Page 8, line 173: The sentence is a little difficult to follow; suggest to write "This error can be considerd as the change in radiance the PS effect leads to, which . . . "

- Page 8, line 175: You write: "Note that this changes negligibly as a function of wavelength". Why? Could the instrument polarization response ($m_{01}$, $m_{02}$) not be strongly wavelength dependent?

- Page 9, line 195: A comma is missing after "respectively"

- Page 9, line 199: You write that $m_{01} = \pm PS$. Is PS a constant here, independent of wavelength. Why?

- Page 10, figure caption: For clarity suggest to write "(blue line and right axis)"

- Page 11, line 220: Parenthesis are missing around "Zoogman et al. (2017)"

- Page 11, line 224: The second occurrence of the word "cases" can be removed

- Page 13, Fig. 8: These plots are not so easy to understand. Could you explain the shaded regions in more detail?

---

## Author Comment (AC1)

Thank you very much for your thoughtful and detailed comments. We believe the changes we will make (as detailed below) will improve the manuscript substantially.

Referee #1 comment responses

The description of the instrument (response) model in Sect. 2.1.2 gives very little details and there is no reference to a more detailed description.

We will modify the first paragraph of this section so that it reads:
"The reference radiance spectra corresponding to the NO2 reference amounts over water, rural and urban scenes were modified by applying the instrument model (for several US locations). The instrument response model was based on the TEMPO design, which consists of of a reflective f/3 Schmidt-form telescope and a spectrometer assembly that utilizes a diffraction grating to form an image on CCD detector arrays (Zoogman et al. (2017)). The simulated radiance was modified by this instrument response model, which sampled the radiance at 0.2 nm wavelength steps with a resolution of 0.6 nm, and applied a PS response.125 The PS response model was not specific to TEMPO as our goal was to understand the range of impacts associated with the ACX polarization requirements. The noise was also applied as defined by the ACX signal-to-noise (SNR) specification. Our instrument parameters from TEMPO were modified by assuming a sampling strategy or integration time modification that brought the noise in line with that specified by ACX. Table 2 shows the parameters included in this model."

Note that a reference was included that details the TEMPO design:
Zoogman 2017 https://doi.org/10.1016/j.jqsrt.2016.05.008

How does polarization influence the measurement?

We will add more intuitive description of the influence of polarization before introducing the formalism: "If the instrument is sensitive to light with a certain polarization, this variation in degree of linear polarization translates to a variation in measured radiance throughout the day. Thus, limiting the PS of the satellite sensor can limit the radiometric uncertainty. "

What kind of spectrally-dependent polarization features may be expected? What knowledge is available from GEMS and TEMPO, instruments that do not have a polarization scrambler? Could a _gure be provided showing polarization sensitivity (PS) vs. wavelength?

The expected polarization sensitivity spectrum will depend on the ACX architecture yet to be developed. Although we used TEMPO-like parameters for assessing noise impacts, we chose to keep the polarization analysis general to give early assessments without constraining to a particular architecture, since this can vary widely depending on the optical components in the optical path. As noted the Mueller Matrix will be used once it is modeled or measured to give a more accurate and complete description.

Minor comments
Page 1, line 5: Given the formulation of names and acronyms of the other instruments, it is more logical to write ". . . and OMI (Ozone Monitoring Instument), . . . "

Agreed. We will modify.

Page 2, line 26: Given the formulation of names and acronyms of the other instruments, it is more logical to write ". . . such as TROPOspheric Monitoring Instrument (TROPOMI), . . . "

Agreed. We will modify.

Page 2, line 27{28: The "GEMS" discussed by Hollingsworth is an entirely di_erent GEMS than the Korean geostationary spectrometer. Please add an appropriate reference for GEMS. The paper of Hollingsworth is more appropriate for line 30.

You are correct. The reference was not appropriate. We will substitute Kim et al 2020 (doi: 10.1175/bams-d-18-0013.1) and move Hollingsworth reference to line 30.

Page 2, line 35: The paper by van Ge_en et al. (2020) describes the TROPOMI NO2 retrieval and is more appropriate in line 26.

We will substitute a different reference that we believe is more relevant  (Crutzen etal. 1979)

Page 2, line 51: Since both GEO and LEO have been spelled out in line 25, the line can read ". . . in both GEO (. . . ) and LEO, though . . . "

Agreed. We will modify.

Page 3, line 62: For clarity please write ". . . its circular polarization through . . . "

We will modify.

Page 4, Sect. 2.1.1: The light scattered by the surface may also become partly polarized. An example is the scattering by a water surface (e.g. principle behind Polaroid sun glasses). Is this taken into account, and could this affect the analysis?

This effect could be taken into account by specifying a bidirectional polarization distribution function. We have not yet implemented such a model in our simulations. These could have some affect – perhaps on the order of 1% difference in the degree of linear polarization. We chose to neglect this effect in the interest of simplicity but will consider including for future works.
(See Maignan et al. (https://doi.org/10.1016/j.rse.2009.07.022 and Litvinov et al. 2011 (https://doi.org/10.1016/j.rse.2010.11.005))

In the conclusion, we will add "In addition, the limited set of surface reflectance types that were used and the directional and polarization surface effects that were neglected, can be included in future work to improve the accuracy of the results."

Page 5, line 110: A space is missing in "shrubs(30%)"

A space will be inserted.

Page 5, Table 1: Please explain the parameters AOD and sigma. Are the aerosols a mix of scattering and absorbing aerosol types?

"Loading" will be replaced with "aerosol optical depth (AOD)" in the text and the acronym will also be defined in the footnote of the Table. We will also add the mean index of refraction for the aerosols to show their absorption properties.

Page 6, line 132: The second occurrence of "all" at the start of the line needs to be removed

Will be removed.

Page 7, line 145: Surely the cloudy radiance $L_{cld}$ is meant here.

The symbol will be changed.

Are these radiances polarized, or unpolarized? Would that influence the result?

The predicted radiance terms are derived from the unpolarized component. This would only affect the result if the instrument were designed to measure the polarization state.

Page 7, line 150: A comma is missing in ". . . cloud radiance fraction, $f_r$, . . . "

We will insert a comma.

Page 8, lines 171{172: The "temperature correction" appears out of nowhere and it is not clear what it refers to. Is it related to the temperature dependence of the $NO_2$ cross sections, which is compensated by a temperature correction term in the AMF calculation in the $NO_2$ retrievals of e.g. OMI and TROPOMI?

Yes. This refers to empirical temperature correction coefficient accounting for the temperature dependence of the NO2 absorption cross-section. We will add this description to the text. "A correction term, $\alpha$, is normally included in the AMF calculation to account for the temperature dependence of the $NO_2$ cross sections, though was neglected here by setting it to one"

Page 8, line 173: The sentence is a little di_cult to follow; suggest to write "This error can be considerd as the change in radiance the PS e_ect leads to, which . . . "

Agreed. We will change to "This error can be considered the effect of a change in detected radiance due to PS, which, in turn, leads to an error in the interpretation of the amount of clouds in the scene."

Page 8, line 175: You write: "Note that this changes negligibly as a function of wavelength". Why? Could the instrument polarization response ($m_{01}$, $m_{02}$) not be strongly wavelength dependent?

The NO2 retrieval errors could indeed vary if the PS changes as a function of wavelength. In this study, having no such information, we assume a constant PS parameters with wavelength. We will clarify "Note that assuming a constant polarization sensitivity over the wavelength range, this error will also change negligibly as a function of wavelength.."

Page 9, line 195: A comma is missing after "respectively"

Will change.

Page 9, line 199: You write that $m_{01} = \_PS$. Is PS a constant here, independent of wavelength. Why?

See the comment above.

Page 10, figure caption: For clarity suggest to write "(blue line and right axis)"

Will change.

Page 11, line 220: Parenthesis are missing around "Zoogman et al. (2017)"

Will change.

Page 11, line 224: The second occurrence of the word "cases" can be removed

Will remove.

Page 13, Fig. 8: These plots are not so easy to understand. Could you explain the shaded regions in more detail?

Will add the sentence "The shading is meant to emphasize the difference between the reference and retrieved amount."

---

## Author Comment (AC2)

Thank you very much for your thorough and thoughtful comments. We believe that in addressing these comments, the manuscript will improve significantly.

**General comments:**

In section 2, several simulation methods are given, which include a multitude of parameters.

In section 3, the results of these simulations are presented. Here, the references to simulation parameters are given by (only) partly repeating these from section 2. At initial reading, this causes difficulty and a lot of back-and-forth reading in relating the results to the simulation parameters. And I still have to guess.

I strongly suggest to make this more structured, by clearly itemizing in section 2, like "simulation A: ...", "simulation B: ..."

and then referring to these cases in section 3

We will delineate the different simulations more clearly by adding a table to section 2 that gives the key parameters (NO2 amount, solar zenith angle/time, scene type, polarization sensitivity, orientation, and locations) for each of the four main simulations. We will then clearly reference these in section 3 in the text and figure captions.

It is shown in the paper that instrument polarisation sensitivity mainly affects the AMF retrieval, not the NO2 slant columns.

In this study, the polarisation sensitivity enters NO2 vertical column through the retrieval of cloud fraction.

But in operational retrievals, also the cloud altitude and(or) cloud optical thickness must be derived.

Especially for cloud retrieval that uses information from polarised radiance (e.g. rotational raman scattering or deep absorption bands of O2) it may be expected that instrument polarisation plays a role. It is understood that the cloud retrieval algorithm for an instrument in initial development is TBD and simulations are premature. Nevertheless this should be mentioned explicitly.

We added the following: "Our simplified retrieval approach may have neglected factors used in operational retrievals that could be affected by instrument PS and

contribute to additional retrieval errors related to estimates of aerosols, surface reflectance, and cloud parameters."

Specific comments:

Figure 2: Surface spectra are shown here from ~400 to 3000+ nm. But the NO2 retrieval window is 420-455 nm and from these figures it is impossible to see any spectral structure there. Please reduce the spectral range of the figure and comment on the surface spectral resolution.

We have added a plot to Figure 2 showing the spectra in the 420-455 nm range.

lines 160-175: here the description becomes confusing/sloppy:

AMF in line 163 seems to be the height-dependent box-AMF which in Eq.(10) should be written as function of *z*.

I assume you mean AMF in line 165.

Yes, we will clarify and specify box AMF

What is the relation between AMF\_tot in Eq.(7) and AMF\_total in Eq.(10)? Meant is probably that Eq.(7) in incorporated in Eq.(10). I suggest to write this out in Eq.(10).

In order to explain the alpha in Eq.(10) I suggest to say beforehand that the formulation follows Kuhlmann 2015.

In line 175, should not  $\partial$ AMF\_tot be  $\partial$ AMF\_total, and is its only dependence on PS through Eq.(8) ? Please rewrite to make that explicit.

We will clarify this section by being more explicit and consistent in our AMF terminology. We have changed the equation 9-11:

- Eq. 9: AMFtotal will replace AMFtot
- Eq 10: The box AMF is shown as a function of z (and referenced in the text), and AMFclr/cld will replace AMFtotal.
- Eq. 11:  $\partial$ (AMFtotal) will replace  $\partial$ (AMFtot)

section 2.2.2:

were the same aerosol parameters as for clear sky used? or no aerosol at all?

There are no aerosols for the cloudy scenes. We will add the bolded text to: " we model Lclr as a Lambertian surface (opaque) with surface reflectivity 0.80 at the effective cloud pressure, assumed here to be equivalent to a cloud at 2 km. Aerosols are not considered for the cloudy scenes, since they would have a negligible impact; the clouds would lie above the tropospheric NO2 and aerosol layer."

it is tacitly assumed that cloudy pixels have unpolarised radiance. Please mention/explain this explicitly.

This is not the case in general. The confusion may be arising from the simplified illustration (Fig. 3) where the polarization response is not being applied to the cloudy scene. This is only true when we start from a clear scene and look at the how the deviation in radiance form the PS changes how this affect the interpretation of the cloud fraction. When we start with an assumed non-zero cloud fraction, the output of the RTM includes a combination of the simulated Stokes from the clear and cloudy scenes (when for instance, we used the GEOS-5 profiles and cloud fractions).

We will add the bold text in the following to clarify this point: "This deviates from the illustration in Fig. 1 (top left), where instead of a clear scene, a mixture of cloudy and clear scene according to the GEOS-5 cloud fraction value is used, **thereby accounting for the radiance polarization state of both clear and cloudy scenes in generating the NO2 retrieval errors**"

line 208: "other retrieval techniques that do not use a spectral fitting approach" should be

"other retrieval techniques that do not use a polynomial correction term in the spectral fitting approach"

**Will change.**

Figure 5b. Why is the standard deviation over Water so much smaller than over Land? Usually the reflectances over water are smaller so S/N should be worse, unless you force S/N to be constant (as suggested in the text). Or is something else the case like an aerosol effect or a spectral surface effect? Please explain.

The water scene had lower retrieval errors for this location. This is not always the case, and the differences in SNR do not always drive the retrieval errors. We have not isolated the cause, but it is likely a combination of SNR, view geometry, and aerosol models that contribute to the retrieval errors.

We will change, "The errors are driven by a combination of the noise and are similar for all scene types" to "The errors are driven by a combination of the SNR, view/solar geometry, surface reflectance spectrum, and aerosol model and are similar for all scene types."

section 3.2 . Confusing: which simulation from section 2.2.2 was used to generate Figure 7 and which one for Figure 9?

Please refer to the previous comment. Adding the table and references to it will hopefully clarify.

See my general comment.

Figure 7 has fixed surface type thus seems to be the second simulation from section 2.2.2. Figure 9 uses GEOS-5 data thus also seems to be the second simulation ??

**The simulation identifier will be added to the caption**

What means " water, rural, urban scene covers CONUS [...] for each surface? use for each grid cell the most abundant type? What means "with a fixed scene type over the CONUS grid" ? use 1 type for all grid cells?

Please provide a bit less condensed description.

Added a sentence to 2.2.2 in bold: "Simulation B quantifies the retrieval impact of scene type, water, rural, and urban scene, over CONUS for a constant reference NO2 profile. The scene types are the same as defined in Table A and are assigned to all pixels in CONUS for each run."

line 220: parenthesis typo in Zoogman reference

**Will fix.**

Retrieval results for Figure 6: Is it correct that these simulations were done with a small (<0.04) cloud fraction? "Cloud radiance fraction" refers to the retrieved result

(doesn't it ?). Please specify which cloud fraction was used in the forward simulation for this figure.

Will add the bolded text to the description of simulation "The simulation ran using 70° solar zenith angle and water scene for all pixels and an instrument PS of 5%, m01 = -0.05, vertical orientation and m02 = 0.05, 45° orientation, **and an initial cloud fraction of zero**." Will also add "**retrieved** cloud fraction" to refer to the result after considering PS effects.

Figure 8: data for NO2 amounts 0f 5.0E+15 and 8.6E+15 are difficult to read in this figure. Is the relative error ("percent error") approximately equal for all three NO2 amounts? Please adjust figure or mention in the text.

The % error will be added in the text. It is non-linear with NO2 amount. We will add: "The dependence on NO2 amount is non-linear; for instance, at 5 % PS for the Seattle, evening case, the retrieval errors for increasing amounts are 0.22 %, 2.6 %, and 6.6 %, respectively.

Retrieval results for Figure 9: are these retrievals with "fixed scene type" as suggested for the scenario with GEOS profiles in line 181? That would not be very realistic. If water is used for the extreme East/West (Atlantic/Pacific) why are the errors so much smaller than in fig 7c? Your text says "The higher cloud fraction decrease the retrieval errors" but also clear scenes at high solar zenith angle have much smaller errors. Is this because of the NO2 amounts? It would be useful to show a figure with NO2 input column.

Yes. Water was used for all pixels as clarified (see above comment for section 2.2.2). Although this isn't too realistic, the lower reflectance types gives the largest values. We used this type, because we are more interested in upper bounds for requirements considerations. We will add the NO2 amount per location in Figure 9 and the bolded text in the following: **"As a result of the cloud fraction and lower NO2 amount**, the maximum NO2 errors found were 0.03 ×1015molecules/cm2 for this day —a negligible value when compared to the TEMPO precision requirement."

---

## Author Response (AR2)

Table 3 has been corrected: 45° orientation is now under Simulation A.